# Multiresolution Kernel Approximation for Gaussian Process Regression

**Yi Ding**[*], **Risi Kondor**[*][†], **Jonathan Eskreis-Winkler**[†]
[*]Department of Computer Science, [†]Department of Statistics
The University of Chicago, Chicago, IL, 60637
{dingy,risi,eskreiswinkler}@uchicago.edu

## Abstract

Gaussian process regression generally does not scale to beyond a few thousands data points without applying some sort of kernel approximation method. Most approximations focus on the high eigenvalue part of the spectrum of the kernel matrix, $K$, which leads to bad performance when the length scale of the kernel is small. In this paper we introduce Multiresolution Kernel Approximation (MKA), the first true broad bandwidth kernel approximation algorithm. Important points about MKA are that it is memory efficient, and it is a direct method, which means that it also makes it easy to approximate $K^{-1}$ and $\det(K)$.

## 1 Introduction

Gaussian Process (GP) regression, and its frequentist cousin, kernel ridge regression, are such natural and canonical algorithms that they have been reinvented many times by different communities under different names. In machine learning, GPs are considered one of the standard methods of Bayesian nonparametric inference [1]. Meanwhile, the same model, under the name Kriging or Gaussian Random Fields, is the de facto standard for modeling a range of natural phenomena from geophyics to biology [2]. One of the most appealing features of GPs is that, ultimately, the algorithm reduces to "just" having to compute the inverse of a kernel matrix, $K$. Unfortunately, this also turns out to be the algorithm's Achilles heel, since in the general case, the complexity of inverting a dense $n \times n$ matrix scales with $O(n^3)$, meaning that when the number of training examples exceeds $10^4 \sim 10^5$, GP inference becomes problematic on virtually any computer[1]. Over the course of the last 15 years, devising approximations to address this problem has become a burgeoning field.

The most common approach is to use one of the so-called Nyström methods [3], which select a small subset $\{x_{i_1}, \ldots, x_{i_m}\}$ of the original training data points as "anchors" and approximate $K$ in the form $K \approx K_{*,I} C K_{*,I}^\top$, where $K_{*,I}$ is the submatrix of $K$ consisting of columns $\{i_1, \ldots, i_m\}$, and $C$ is a matrix such as the pseudo-inverse of $K_{I,I}$. Nyström methods often work well in practice and have a mature literature offering strong theoretical guarantees. Still, Nyström is inherently a global low rank approximation, and, as pointed out in [4], a priori there is no reason to believe that $K$ should be well approximable by a low rank matrix: for example, in the case of the popular Gaussian kernel $k(x, x') = \exp(-(x - x')^2/(2\ell^2))$, as $\ell$ decreases and the kernel becomes more and more "local" the number of significant eigenvalues quickly increases. This observation has motivated alternative types of approximations, including local, hierarchical and distributed ones (see Section 2). In certain contexts involving translation invariant kernels yet other strategies may be applicable [5], but these are beyond the scope of the present paper.

In this paper we present a new kernel approximation method, Multiresolution Kernel Approximation (MKA), which is inspired by a combination of ideas from hierarchical matrix decomposition

algorithms and multiresolution analysis. Some of the important features of MKA are that (a) it is a broad spectrum algorithm that approximates the entire kernel matrix $K$, not just its top eigenvectors, and (b) it is a so-called "direct" method, i.e., it yields explicit approximations to $K^{-1}$ and $\det(K)$.

**Notations.** We define $[n] = \{1, 2, \ldots, n\}$. Given a matrix $A$, and a tuple $I = (i_1, \ldots, i_r)$, $A_{I,*}$ will denote the submatrix of $A$ formed of rows indexed by $i_1, \ldots, i_r$, similarly $A_{*,J}$ will denote the submatrix formed of columns indexed by $j_1, \ldots, j_p$, and $A_{I,J}$ will denote the submatrix at the intersection of rows $i_1, \ldots, i_r$ and columns $j_1, \ldots, j_p$. We extend these notations to the case when $I$ and $J$ are sets in the obvious way. If $A$ is a blocked matrix then $[\![A]\!]_{i,j}$ will denote its $(i, j)$ block.

## 2 Local vs. global kernel approximation

Recall that a Gaussian Process (GP) on a space $\mathcal{X}$ is a prior over functions $f \colon \mathcal{X} \to \mathbb{R}$ defined by a mean function $\mu(x) = \mathbb{E}[f(x)]$, and covariance function $k(x, x') = \mathrm{Cov}(f(x), f(x'))$. Using the most elementary model $y_i = f(x_i) + \epsilon$ where $\epsilon \sim \mathcal{N}(0, \sigma^2)$ and $\sigma^2$ is a noise parameter, given training data $\{(x_1, y_1), \ldots, (x_n, y_n)\}$, the posterior is also a GP, with mean $\mu'(x) = \mu(x) + \boldsymbol{k}_x^\top (K + \sigma^2 I)^{-1} \boldsymbol{y}$, where $\boldsymbol{k}_x = (k(x, x_1), \ldots, k(x, x_n))$, $\boldsymbol{y} = (y_1, \ldots, y_n)$, and covariance

$$k'(x, x') = k(x, x') - \boldsymbol{k}_{x'}^\top (K + \sigma^2 I)^{-1} \boldsymbol{k}_x. \tag{1}$$

Thus (here and in the following assuming $\mu = 0$ for simplicity), the maximum a posteriori (MAP) estimate of $f$ is

$$\widehat{f}(x) = \boldsymbol{k}_x^\top (K + \sigma^2 I)^{-1} \boldsymbol{y}. \tag{2}$$

Ridge regression, which is the frequentist analog of GP regression, yields the same formula, but regards $\widehat{f}$ as the solution to a regularized risk minimization problem over a Hilbert space $\mathcal{H}$ induced by $k$. We will use "GP" as the generic term to refer to both Bayesian GPs and ridge regression. Letting $K' = (K + \sigma^2 I)$, virtually all GP approximation approaches focus on trying to approximate the (augmented) kernel matrix $K'$ in such a way so as to make inverting it, solving $K'\boldsymbol{y} = \boldsymbol{\alpha}$ or computing $\det(K')$ easier. For the sake of simplicity in the following we will actually discuss approximating $K$, since adding the diagonal term usually doesn't make the problem any more challenging.

### 2.1 Global low rank methods

As in other kernel methods, intuitively, $K_{i,j} = k(x_i, x_j)$ encodes the degree of similarity or closeness between the two points $x_i$ and $x_j$ as it relates to the degree of correlation/similarity between the value of $f$ at $x_i$ and at $x_j$. Given that $k$ is often conceived of as a smooth, slowly varying function, one very natural idea is to take a smaller set $\{x_{i_1}, \ldots, x_{i_m}\}$ of "landmark points" or "pseudo-inputs" and approximate $k(x, x')$ in terms of the similarity of $x$ to each of the landmarks, the relationship of the landmarks to each other, and the similarity of the landmarks to $x'$. Mathematically,

$$k(x, x') \approx \sum_{s=1}^{m} \sum_{j=1}^{m} k(x, x_{i_s}) \, c_{i_s, i_j} \, k(x_{i_j}, x'),$$

which, assuming that $\{x_{i_1}, \ldots, x_{i_m}\}$ is a subset of the original point set $\{x_1, \ldots, x_n\}$, amounts to an approximation of the form $K \approx K_{*,I} C K_{*,I}^\top$, with $I = \{i_1, \ldots, i_m\}$. The canonical choice for $C$ is $C = W^+$, where $W = K_{I,I}$, and $W^+$ denotes the Moore-Penrose pseudoinverse of $W$. The resulting approximation

$$K \approx K_{*,I} W^+ K_{*,I}^\top, \tag{3}$$

is known as the Nyström approximation, because it is analogous to the so-called Nyström extension used to extrapolate continuous operators from a finite number of quadrature points. Clearly, the choice of $I$ is critical for a good quality approximation. Starting with the pioneering papers [6, 3, 7], over the course of the last 15 years a sequence of different sampling strategies have been developed for obtaining $I$, several with rigorous approximation bounds [8, 9, 10, 11]. Further variations include the ensemble Nyström method [12] and the modified Nyström method [13].

Nyström methods have the advantage of being relatively simple, and having reliable performance bounds. A fundamental limitation, however, is that the approximation (3) is inherently low rank. As pointed out in [4], there is no reason to believe that kernel matrices in general should be close to low rank. An even more fundamental issue, which is less often discussed in the literature, relates to the

specific form of (2). The appearance of $K'^{-1}$ in this formula suggests that it is the *low* eigenvalue eigenvectors of $K'$ that should dominate the result of GP regression. On the other hand, multiplying the matrix by $\boldsymbol{k}_x$ largely cancels this effect, since $\boldsymbol{k}_x$ is effectively a row of a kernel matrix similar to $K'$, and will likely concentrate most weight on the *high* eigenvalue eigenvectors. Therefore, ultimately, it is not $K'$ itself, but the relationship between the eigenvectors of $K'$ and the data vector $\boldsymbol{y}$ that determines which part of the spectrum of $K'$ the result of GP regression is most sensitive to.

Once again, intuition about the kernel helps clarify this point. In a setting where the function that we are regressing is smooth, and correspondingly, the kernel has a large length scale parameter, it is the global, long range relationships between data points that dominate GP regression, and that can indeed be well approximated by the landmark point method. In terms of the linear algebra, the spectral expansion of $K'$ is dominated by a few large eigenvalue eigenvectors, we will call this the "PCA-like" scenario. In contrast, in situations where $f$ varies more rapidly, a shorter lengthscale kernel is called for, local relationships between nearby points become more important, which the landmark point method is less well suited to capture. We call this the "$k$–nearest neighbor type" scenario. In reality, most non-trivial GP regression problems fall somewhere in between the above two extremes. In high dimensions data points tend to be all almost equally far from each other anyway, limiting the applicability of simple geometric interpretations. Nonetheless, the two scenarios are an illustration of the general point that one of the key challenges in large scale machine learning is integrating information from both local and global scales.

## 2.2   Local and hierarchical low rank methods

Realizing the limitations of the low rank approach, local kernel approximation methods have also started appearing in the literature. Broadly, these algorithms: (1) first cluster the rows/columns of $K$ with some appropriate fast clustering method, e.g., METIS [14] or GRACLUS [15] and block $K$ accordingly; (2) compute a low rank, but relatively high accuracy, approximation $[\![K]\!]_{i,i} \approx U_i \Sigma_i U_i^\top$ to each diagonal block of $K$; (3) use the $\{U_i\}$ bases to compute possibly coarser approximations to the $[\![K]\!]_{i,j}$ off diagonal blocks. This idea appears in its purest form in [16], and is refined in [4] in a way that avoids having to form all rows/columns of the off-diagonal blocks in the first place. Recently, [17] proposed a related approach, where all the blocks in a given row share the same row basis but have different column bases. A major advantage of local approaches is that they are inherently parallelizable. The clustering itself, however, is a delicate, and sometimes not very robust component of these methods. In fact, divide-and-conquer type algorithms such as [18] and [19] can also be included in the same category, even though in these cases the blocking is usually random.

A natural extension of the blocking idea would be to apply the divide-and-conquer approach recursively, at multiple different scales. Geometrically, this is similar to recent multiresolution data analysis approaches such as [20]. In fact, hierarchical matrix approximations, including HODLR matrices, $\mathcal{H}$–matrices [21], $\mathcal{H}^2$–matrices [22] and HSS matrices [23] are very popular in the numerical analysis literature. While the exact details vary, each of these methods imposes a specific type of block structure on the matrix and forces the off-diagonal blocks to be low rank (Figure 1 in the Supplement). Intuitively, nearby clusters interact in a richer way, but as we move farther away, data can be aggregated more and more coarsely, just as in the fast multipole method [24].

We know of only two applications of the hierarchical matrix methodology to kernel approximation: Börm and Garcke's $\mathcal{H}^2$ matrix approach [25] and O'Neil et al.'s HODLR method [26]. The advantage of $\mathcal{H}^2$ matrices is their more intricate structure, allowing relatively tight interactions between neighboring clusters even when the two clusters are not siblings in the tree (e.g. blocks 8 and 9 in Figure 1c in the Supplement). However, the $\mathcal{H}^2$ format does not directly help with inverting $K$ or computing its determinant: it is merely a memory-efficient way of storing $K$ and performing matrix/vector multiplies *inside* an iterative method. HODLR matrices have a simpler structure, but admit a factorization that makes it possible to directly compute both the inverse and the determinant of the approximated matrix in just $O(n \log n)$ time.

The reason that hierarchical matrix approximations have not become more popular in machine learning so far is that in the case of high dimensional, unstructured data, finding the way to organize $\{x_1, \ldots, x_n\}$ into a single hierarchy is much more challenging than in the setting of regularly spaced points in $\mathbb{R}^2$ or $\mathbb{R}^3$, where these methods originate:  1. Hierarchical matrices require making hard assignments of data points to clusters, since the block structure at each level corresponds to partitioning the rows/columns of the original matrix.  2. The hierarchy must form a single tree, which

puts deep divisions between clusters whose closest common ancestor is high up in the tree. 3. Finding the hierarchy in the first place is by no means trivial. Most works use a top-down strategy which defeats the inherent parallelism of the matrix structure, and the actual algorithm used (kd-trees) is known to be problematic in high dimensions [27].

## 3 Multiresolution Kernel Approximation

Our goal in this paper is to develop a data adapted multiscale kernel matrix approximation method, **Multiresolution Kernel Approximation (MKA)**, that reflects the "distant clusters only interact in a low rank fashion" insight of the fast multipole method, but is considerably more flexible than existing hierarchical matrix decompositions. The basic building blocks of MKA are local factorizations of a specific form, which we call core-diagonal compression.

**Definition 1** *We say that a matrix $H$ is* **c–core-diagonal** *if $H_{i,j} = 0$ unless either $i, j \leq c$ or $i = j$.*

**Definition 2** *A* **c–core-diagonal compression** *of a symmetric matrix $A \in \mathbb{R}^{m \times m}$ is an approximation of the form*

$$A \approx Q^\top H\, Q = \left( \blacksquare \right) \left( \blacksquare \diagdown \right) \left( \blacksquare \right), \tag{4}$$

*where $Q$ is orthogonal and $H$ is c–core-diagonal.*

Core-diagonal compression is to be contrasted with rank $c$ sketching, where $H$ would just have the $c \times c$ block, without the rest of the diagonal. From our multiresolution inspired point of view, however, the purpose of (4) is not just to sketch $A$, but to also to split $\mathbb{R}^m$ into the direct sum of two subspaces: (a) the "detail space", spanned by the last $n-c$ rows of $Q$, responsible for capturing purely local interactions in $A$ and (b) the "scaling space", spanned by the first $c$ rows, capturing the overall structure of $A$ and its relationship to other diagonal blocks.

Hierarchical matrix methods apply low rank decompositions to many blocks of $K$ in parallel, at different scales. MKA works similarly, by applying core-diagonal compressions. Specifically, the algorithm proceeds by taking $K$ through a sequence of transformations $K = K_0 \mapsto K_1 \mapsto \ldots \mapsto K_s$, called stages. In the first stage

1. Similar to other local methods, MKA first uses a fast clustering method to cluster the rows/columns of $K_0$ into clusters $\mathcal{C}_1^1, \ldots, \mathcal{C}_{p_1}^1$. Using the corresponding permutation matrix $C_1$ (which maps the elements of the first cluster to $(1, 2, \ldots |\mathcal{C}_1^1|)$, the elements of the second cluster to $(|\mathcal{C}_1^1| + 1, \ldots, |\mathcal{C}_1^1| + |\mathcal{C}_2^1|)$, and so on) we form a blocked matrix $\overline{K_0} = C_1 K_0 C_1^\top$, where $[\![\overline{K_0}]\!]_{i,j} = K_{\mathcal{C}_i^1, \mathcal{C}_j^1}$.

2. Each diagonal block of $\overline{K_0}$ is independently core-diagonally compressed as in (4) to yield
$$H_i^1 = \left( Q_i^1\, [\![\overline{K_0}]\!]_{i,i}\, (Q_i^1)^\top \right)_{CD(c_i^1)} \tag{5}$$
where $CD(c_i^1)$ in the index stands for truncation to $c_i^1$–core-diagonal form.

3. The $Q_i^1$ local rotations are assembled into a single large orthogonal matrix $\overline{Q_1} = \bigoplus_i Q_i^1$ and applied to the full matrix to give $\overline{H_1} = \overline{Q_1}\, \overline{K_0}\, \overline{Q_1}^\top$.

4. The rows/columns of $\overline{H_1}$ are rearranged by applying a permutation $P_1$ that maps the core part of each block to one of the first $c_1 := c_1^1 + \ldots c_{p_1}^1$ coordinates, and the diagonal part to the rest, giving $H_1^{\text{pre}} = P_1\, \overline{H_1}\, P_1^\top$.

5. Finally, $H_1^{\text{pre}}$ is truncated into the core-diagonal form $H_1 = K_1 \oplus D_1$, where $K_1 \in \mathbb{R}^{c_1 \times c_1}$ is dense, while $D_1$ is diagonal. Effectively, $K_1$ is a compressed version of $K_0$, while $D_1$ is formed by concatenating the diagonal parts of each of the $H_i^1$ matrices. Together, this gives a global core-diagonal compression
$$K_0 \approx \underbrace{C_1^\top \overline{Q_1}^\top P_1^\top}_{\mathcal{Q}_1^\top} (K_1 \oplus D_1) \underbrace{P_1 \overline{Q_1} C_1}_{\mathcal{Q}_1}$$
of the entire original matrix $K_0$.

The second and further stages of MKA consist of applying the above five steps to $K_1, K_2, \ldots, K_{s-1}$ in turn, so ultimately the algorithm yields a kernel approximation $\tilde{K}$ which has a telescoping form

$$\tilde{K} \approx \mathcal{Q}_1^\top (\mathcal{Q}_2^\top (\ldots \mathcal{Q}_s^\top (K_s \oplus D_s) \mathcal{Q}_s \ldots \oplus D_2) \mathcal{Q}_2 \oplus D_1) \mathcal{Q}_1 \tag{6}$$

The pseudocode of the full algorithm is in the Supplementary Material.

MKA is really a meta-algorithm, in the sense that it can be used in conjunction with different core-diagonal compressors. The main requirements on the compressor are that (a) the core of $H$ should capture the dominant part of $A$, in particular the subspace that most strongly interacts with other blocks, (b) the first $c$ rows of $Q$ should be as sparse as possible. We consider two alternatives.

**Augmented Sparse PCA (SPCA).** Sparse PCA algorithms explicitly set out to find a set of vectors $\{\boldsymbol{v}_1, \ldots, \boldsymbol{v}_c\}$ so as to maximize $\|V^\top A V\|_{\mathrm{Frob}}$, where $V = [\boldsymbol{v}_1, \ldots, \boldsymbol{v}_c]$, while constraining each vector to be as sparse as possible [28]. While not all SPCAs guarantee orthogonality, this can be enforced a posteriori via e.g., QR factorization, yielding $Q_{\mathrm{sc}}$, the top $c$ rows of $Q$ in (4). Letting $U$ be a basis for the complementary subspace, the optimal choice for the bottom $m - c$ rows in terms of minimizing Frobenius norm error of the compression is $Q_{\mathrm{wlet}} = U\hat{O}$, where

$$\hat{O} = \operatorname*{argmax}_{O^\top O = I} \| \operatorname{diag}(O^\top U^\top A \, UO)\|,$$

the solution to which is of course given by the eigenvectors of $U^\top A U$. The main drawback of the SPCA approach is its computational cost: depending on the algorithm, the complexity of SPCA scales with $m^3$ or worse [29, 30].

**Multiresolution Matrix Factorization (MMF)** MMF is a recently introduced matrix factorization algorithm motivated by similar multiresolution ideas as the present work, but applied at the level of individual matrix entries rather than at the level of matrix blocks [31]. Specifically, MMF yields a factorization of the form

$$A \approx \underbrace{q_1^\top \ldots q_L^\top}_{Q^\top} H \underbrace{q_L \ldots q_1}_{Q},$$

where, in the simplest case, the $q_i$'s are just Givens rotations. Typically, the number of rotations in MMF is $O(m)$. MMF is efficient to compute, and sparsity is guaranteed by the sparsity of the individual $q_i$'s and the structure of the algorithm. Hence, MMF has complementary strengths to SPCA: it comes with strong bounds on sparsity and computation time, but the quality of the scaling/wavelet space split that it produces is less well controlled.

**Remarks.** We make a few remarks about MKA. 1. Typically, low rank approximations reduce dimensionality quite aggressively. In contrast, in core-diagonal compression $c$ is often on the order of $m/2$, leading to "gentler" and more faithful, kernel approximations. 2. In hierarchical matrix methods, the block structure of the matrix is defined by a single tree, which, as discussed above, is potentially problematic. In contrast, by virtue of reclustering the rows/columns of $K_\ell$ before every stage, MKA affords a more flexible factorization. In fact, beyond the first stage, it is not even individual data points that MKA clusters, but subspaces defined by the earlier local compressions. 3. While $C_\ell$ and $P_\ell$ are presented as explicit permutations, they really just correspond to different ways of blocking $K_s$, which is done implicitly in practice with relatively little overhead. 4. Step 3 of the algorithm is critical, because it extends the core-diagonal splits found in the diagonal blocks of the matrix to the off-diagonal blocks. Essentially the same is done in [4] and [17]. This operation reflects a structural assumption about $K$, namely that the same bases that pick out the dominant parts of the diagonal blocks (composed of the first $c_i^\ell$ rows of the $Q_i^\ell$ rotations) are also good for compressing the off-diagonal blocks. In the hierarchical matrix literature, for the case of specific kernels sampled in specific ways in low dimensions, it is possible to *prove* such statements. In our high dimensional and less structured setting, deriving analytical results is much more challenging. 5. MKA is an inherently bottom-up algorithm, including the clustering, thus it is naturally parallelizable and can be implemented in a distributed environment. 6. The hierarchical structure of MKA is similar to that of the parallel version of MMF (pMMF) [32], but the way that the compressions are calculated is different (pMMF tries to minimize an objective that relates to the entire matrix).

## 4    Complexity and application to GPs

For MKA to be effective for large scale GP regression, it must be possible to compute the factorization fast. In addition, the resulting approximation $\tilde{K}$ must be symmetric positive semi-definite (spsd) (MEKA, for example, fails to fulfill this [4]). We say that a matrix approximation algorithm $A \mapsto \tilde{A}$ is **spsd preserving** if $\tilde{A}$ is spsd whenever $A$ is. It is clear from its form that the Nyström approximation is spsd preserving , so is augmented SPCA compression. MMF has different variants, but the core part of $H$ is always derived by conjugating $A$ by rotations, while the diagonal elements are guaranteed to be positive, therefore MMF is spsd preserving as well.

**Proposition 1** *If the individual core-diagonal compressions in MKA are spsd preserving, then the entire algorithm is spsd perserving.*

The complexity of MKA depends on the complexity of the local compressions. Next, we assume that to leading order in $m$ this cost is bounded by $c_{\text{comp}} m^{\alpha_{\text{comp}}}$ (with $\alpha_{\text{comp}} \geq 1$) and that each row of the $Q$ matrix that is produced is $c_{\text{sp}}$–sparse. We assume that the MKA has $s$ stages, the size of the final $K_s$ "core matrix" is $d_{\text{core}} \times d_{\text{core}}$, and that the size of the largest cluster is $m_{\text{max}}$. We assmue that the maximum number of clusters in any stage is $b_{\text{max}}$ and that the clustering is close to balanced in the sense that that $b_{\text{max}} = \theta(n/m_{\text{max}})$ with a small constant. We ignore the cost of the clustering algorithm, which varies, but usually scales linearly in $snb_{\text{max}}$. We also ignore the cost of permuting the rows/columns of $K_\ell$, since this is a memory bound operation that can be virtualized away. The following results are to leading order in $m_{\text{max}}$ and are similar to those in [32] for parallel MMF.

**Proposition 2** *With the above notations, the number of operations needed to compute the MKA of an $n \times n$ matrix is upper bounded by $2sc_{sp}n^2 + sc_{comp}m_{max}^{\alpha_{comp}-1}n$. Assuming $b_{max}$–fold parallelism, this complexity reduces to $2sc_{sp}n^2/b_{max} + sc_{comp}m_{max}^{\alpha_{comp}}$.*

The memory cost of MKA is just the cost of storing the various matrices appearing in (6). We only include the number of non-zero reals that need to be stored and not indices, etc..

**Proposition 3** *The storage complexity of MKA is upper bounded by $(sc_{sp}+1)n + d_{core}^2$.*

Rather than the general case, it is more informative to focus on MMF based MKA, which is what we use in our experiments. We consider the simplest case of MMF, referred to as "greedy-Jacobi" MMF, in which each of the $q_i$ elementary rotations is a Given rotation. An additional parameter of this algorithm is the compression ratio $\gamma$, which in our notation is equal to $c/n$. Some of the special features of this type of core-diagonal compression are:

(a) While any given row of the rotation $Q$ produced by the algorithm is not guaranteed to be sparse, $Q$ will be the product of exactly $\lfloor(1-\gamma)m\rfloor$ Givens rotations.
(b) The leading term in the cost is the $m^3$ cost of computing $A^\top A$, but this is a BLAS operation, so it is fast.
(c) Once $A^\top A$ has been computed, the cost of the rest of the compression scales with $m^2$.

Together, these features result in very fast core-diagonal compressions and a very compact representation of the kernel matrix.

**Proposition 4** *The complexity of computing the MMF-based MKA of an $n \times n$ dense matrix is upper bounded by $4sn^2 + sm_{max}^2n$, where $s = \log(d_{core}/n)/(\log \gamma)$. Assuming $b_{max}$–fold parallelism, this is reduced to $4snm_{max} + m_{max}^3$.*

**Proposition 5** *The storage complexity of MMF-based MKA is upper bounded by $(2s+1)n + d_{core}^2$.*
Typically, $d_{\text{core}} = O(1)$. Note that this implies $O(n \log n)$ storage complexity, which is similar to Nyström approximations with very low rank. Finally, we have the following results that are critical for using MKA in GPs.

**Proposition 6** *Given an approximate kernel $\tilde{K}$ in MMF-based MKA form (6), and a vector $\boldsymbol{z} \in \mathbb{R}^n$ the product $\tilde{K}\boldsymbol{z}$ can be computed in $4sn + d_{core}^2$ operations. With $b_{max}$–fold parallelism, this is reduced to $4sm_{max} + d_{core}^2$.*

**Proposition 7** *Given an approximate kernel $\tilde{K}$ in (MMF or SPCA-based) MKA form, the MKA form of $\tilde{K}^\alpha$ for any $\alpha$ can be computed in $O(n + d_{core}^3)$ operations. The complexity of computing the matrix exponential $\exp(\beta\tilde{K})$ for any $\beta$ in MKA form and the complexity of computing $\det(\tilde{K})$ are also $O(n + d_{core}^3)$.*

## 4.1 MKA–GPs and MKA Ridge Regression

The most direct way of applying MKA to speed up GP regression (or ridge regression) is simply using it to approximate the augmented kernel matrix $K' = (K + \sigma^2 I)$ and then inverting this approximation using Proposition 7 (with $\alpha = -1$). Note that the resulting $\tilde{K}'^{-1}$ never needs to be evaluated fully, in matrix form. Instead, in equations such as (2), the matrix-vector product $\tilde{K}'^{-1}\boldsymbol{y}$ can be computed in "matrix-free" form by cascading $\boldsymbol{y}$ through the analog of (6). Assuming that $d_{\text{core}} \ll n$ and $m_{\text{max}}$ is not too large, the serial complexity of each stage of this computation scales with at most $n^2$, which is the same as the complexity of computing $K$ in the first place.

One potential issue with the above approach however is that because MKA involves repeated truncation of the $H_j^{\text{pre}}$ matrices, $\tilde{K}'$ will be a biased approximation to $K$, therefore expressions such as (2)

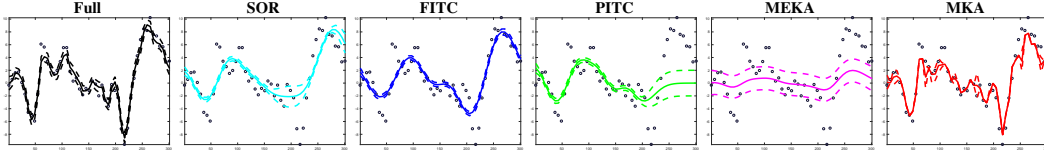

Figure 1: Snelson's 1D example: ground truth (black circles); prediction mean (solid line curves); one standard deviation in prediction uncertainty (dashed line curves).

Table 1: Regression Results with $k$ to be # pseudo-inputs/$d_{\text{core}}$ : SMSE(MNLP)

| Method | k | Full | SOR | FITC | PITC | MEKA | MKA |
|---|---|---|---|---|---|---|---|
| housing | 16 | $0.36(-0.32)$ | $0.93(-0.03)$ | $0.91(-0.04)$ | $0.96(-0.02)$ | $0.85(-0.08)$ | $\mathbf{0.52(-0.32)}$ |
| rupture | 16 | $0.17(-0.89)$ | $0.94(-0.04)$ | $0.96(-0.04)$ | $0.93(-0.05)$ | $0.46(-0.18)$ | $\mathbf{0.32(-0.54)}$ |
| wine | 32 | $0.59(-0.33)$ | $0.86(-0.07)$ | $0.84(-0.03)$ | $0.87(-0.07)$ | $0.97(-0.12)$ | $\mathbf{0.70(-0.23)}$ |
| pageblocks | 32 | $0.44(-1.10)$ | $0.86(-0.57)$ | $0.81(-0.78)$ | $0.86(-0.72)$ | $0.96(-0.10)$ | $\mathbf{0.63(-0.85)}$ |
| compAct | 32 | $0.58(-0.66)$ | $0.88(-0.13)$ | $0.91(-0.08)$ | $0.88(-0.14)$ | $0.75(-0.21)$ | $\mathbf{0.60(-0.32)}$ |
| pendigit | 64 | $0.15(-0.73)$ | $0.65(-0.19)$ | $0.70(-0.17)$ | $0.71(-0.17)$ | $0.53(-0.29)$ | $\mathbf{0.30(-0.42)}$ |

which mix an approximate $K'$ with an exact $\boldsymbol{k}_x$ will exhibit some systematic bias. In Nyström type methods (specifically, the so-called Subset of Regressors and Deterministic Training Conditional GP approximations) this problem is addressed by replacing $\boldsymbol{k}_x$ with its own Nyström approximation, $\hat{\boldsymbol{k}}_x = K_{*,I}W^+\boldsymbol{k}_x^I$, where $[\hat{k}_x^I]_j = k(x, x_{i_j})$. Although $\hat{K}' = K_{*,I}W^+K_{*,I}^\top + \sigma^2 I$ is a large matrix, expressions such as $\hat{\boldsymbol{k}}_x^\top \hat{K}'^{-1}$ can nonetheless be efficiently evaluated by using a variant of the Sherman–Morrison–Woodbury identity and the fact that $W$ is low rank (see [33]).

The same approach cannot be applied to MKA because $\tilde{K}$ is not low rank. Assuming that the testing set $\{x_1, \ldots, x_p\}$ is known at training time, however, instead of approximating $K$ or $K'$, we compute the MKA approximation of the joint train/test kernel matrix

$$\mathcal{K} = \left( \begin{array}{c|c} K & K_* \\ \hline K_*^\top & K_{\text{test}} \end{array} \right) \qquad \text{where} \qquad \begin{array}{l} K_{i,j} = k(x_i, x_j) + \sigma^2 \\ [K_*]_{i,j} = k(x_i, x'_j) \\ [K_{\text{test}}]_{i,j} = k(x'_i, x'_j). \end{array}$$

Writing $\mathcal{K}^{-1}$ in blocked form

$$\tilde{\mathcal{K}}^{-1} = \left( \begin{array}{c|c} A & B \\ \hline C & D \end{array} \right),$$

and taking the Schur complement of $D$ now recovers an alternative approximation $\check{K}^{-1} = A - BD^{-1}C$ to $K^{-1}$ which is consistent with the off-diagonal block $K^*$ leading to our final MKA–GP formula $\widehat{\boldsymbol{f}} = K_*^\top \check{K}^{-1}\boldsymbol{y}$, where $\widehat{\boldsymbol{f}} = (\widehat{f}(x'_1), \ldots, \widehat{f}(x'_p))^\top$. While conceptually this is somewhat more involved than naively estimating $K'$, assuming $p \ll n$, the cost of inverting $D$ is negligible, and the overall serial complexity of the algorithm remains $(n + p)^2$.

In certain GP applications, the $O(n^2)$ cost of writing down the kernel matrix is already forbidding. The one circumstance under which MKA can get around this problem is when the kernel matrix is a matrix polynomial in a sparse matrix $L$, which is most notably for diffusion kernels and certain other graph kernels. Specifically in the case of MMF-based MKA, since the computational cost is dominated by computing local "Gram matrices" $A^\top A$, when $L$ is sparse, and this sparsity is retained from one compression to another, the MKA of sparse matrices can be computed very fast. In the case of graph Laplacians, empirically, the complexity is close to linear in $n$. By Proposition 7, the diffusion kernel and certain other graph kernels can also be approximated in about $O(n \log n)$ time.

## 5 Experiments

We compare MKA to five other methods: 1. **Full**: the full GP regression using Cholesky factorization [1]. 2. **SOR**: the Subset of Regressors method (also equivalent to DTC in mean) [1]. 3. **FITC**: the Fully Independent Training Conditional approximation, also called Sparse Gaussian Processes using Pseudo-inputs [34]. 4. **PITC**: the Partially Independent Training Conditional approximation method (also equivalent to PTC in mean) [33]. 5. **MEKA**: the Memory Efficient Kernel Approximation method [4]. The KISS-GP [35] and other interpolation based methods are not discussed in this paper, because, we believe, they mostly only apply to low dimensional settings. We used custom Matlab implementations [1] for Full, SOR, FITC, and PITC. We used the Matlab codes provided by

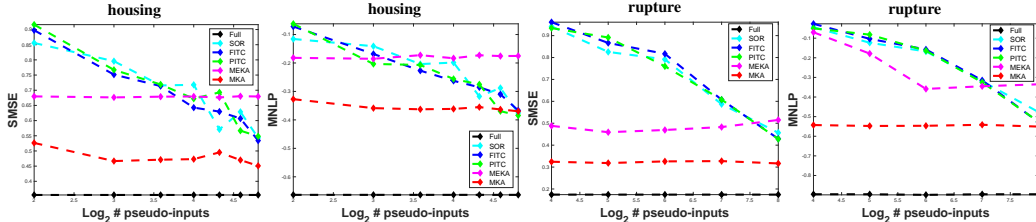

Figure 2: SMSE and MNLP as a function of the number of pseudo-inputs/$d_{\text{core}}$ on two datasets. In the given range MKA clearly outperforms the other methods in both error measures.

the author for MEKA. Our algorithm MKA was implemented in C++ with the Matlab interface. To get an approximately fair comparison, we set $d_{\text{core}}$ in MKA to be the number of pseudo-inputs. The parallel MMF algorithm was used as the compressor due to its computational strength [32]. The Gaussian kernel is used for all experiments with one length scale for all input dimensions.

**Qualitative results.** We show the qualitative behavior of each method on the 1D toy dataset from [34]. We sampled the ground truth from a Gaussian processes with length scale $\ell = 0.5$ and number of pseudo-inputs ($d_{\text{core}}$) is 10. We applied cross-validation to select the parameters for each method to fit the data. Figure 1 shows that MKA fits the data almost as well as the Full GP does. In terms of the other approximate methods, although their fit to the data is smoother, this is to the detriment of capturing the local structure of the underlying data, which verifies MKA's ability to capture the entire spectrum of the kernel matrix, not just its top eigenvectors.

**Real data.** We tested the efficacy of GP regression on real-world datasets. The data are normalized to mean zero and variance one. We randomly selected 10% of each dataset to be used as a test set. On the other 90% we did five-fold cross validation to learn the length scale and noise parameter for each method and the regression results were averaged over repeating this setting five times. All experiments were ran on a 3.4GHz 8 core machine with 8GB of memory. Two distinct error measures are used to assess performance: (a) standardized mean square error (SMSE), $\frac{1}{n}\sum_{t=1}^{n}(\hat{y}_t - y_t)^2/\hat{\sigma}_\star^2$, where $\hat{\sigma}_\star^2$ is the variance of test outputs, and (2) mean negative log probability (MNLP) $\frac{1}{n}\sum_{t=1}^{n}\left((\hat{y}_t - y_t)^2/\hat{\sigma}_\star^2 + \log\hat{\sigma}_\star^2 + \log 2\pi\right)$, each of which corresponds to the predictive mean and variance in error assessment. From Table 1, we are competitive in both error measures when the number of pseudo-inputs ($d_{\text{core}}$) is small, which reveals low-rank methods' inability in capturing the local structure of the data. We also illustrate the performance sensitivity by varying the number of pseudo-inputs on selected datasets. In Figure 2, for the interval of pseudo-inputs considered, MKA's performance is robust to $d_{\text{core}}$, while low-rank based methods' performance changes rapidly, which shows MKA's ability to achieve good regression results even with a crucial compression level. The Supplementary Material gives a more detailed discussion of the datasets and experiments.

# 6   Conclusions

In this paper we made the case that whether a learning problem is low rank or not depends on the nature of the data rather than just the spectral properties of the kernel matrix $K$. This is easiest to see in the case of Gaussian Processes, which is the algorithm that we focused on in this paper, but it is also true more generally. Most existing sketching algorithms used in GP regression force low rank structure on $K$, either globally, or at the block level. When the nature of the problem is indeed low rank, this might actually act as an additional regularizer and improve performance. When the data does not have low rank structure, however, low rank approximations will fail. Inspired by recent work on multiresolution factorizations, we proposed a mulitresolution meta-algorithm, MKA, for approximating kernel matrices, which assumes that the *interaction* between distant clusters is low rank, while avoiding forcing a low rank structure of the data locally, at any scale. Importantly, MKA allows fast direct calculations of the inverse of the kernel matrix and its determinant, which are almost always the computational bottlenecks in GP problems.

### Acknowledgements

This work was completed in part with resources provided by the University of Chicago Research Computing Center. The authors wish to thank Michael Stein for helpful suggestions.

## Footnotes

[1] In the limited case of evaluating a GP with a fixed Gram matrix on a single training set, GP inference reduces to solving a linear system in $K$, which scales better with $n$, but might be problematic behavior when the condition number of $K$ is large.

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
