[Supplementary Material]

# Multiresolution Kernel Approximation for Gaussian Process Regression: Supplementary Material

**Yi Ding[*], Risi Kondor[*][†], Jonathan Eskreis-Winkler[†]**
[*]Department of Computer Science, [†]Department of Statistics
The University of Chicago, Chicago, IL, 60637
{dingy, risi, eskreiswinkler}@uchicago.edu

## 1 Block structure for different hierarchical matrix approximations

Figure 1:  (a) In a simple blocked low rank approximation the diagonal blocks are dense (gray), whereas the off-diagonal blocks are low rank. (b) In an HODLR matrix the low rank off-diagonal blocks form a hierarchical structure leading to a much more compact representation. (c) $\mathcal{H}^2$ matrices are a refinement of this idea.

## 2 Proofs

**Proof of Proposition 1.** Assume that $K_{\ell-1}$ is spsd. Stage $\ell$ transforms $K_{\ell-1}$ to $H_\ell = K_\ell \oplus D_\ell$. Here $K_\ell$ is a submatrix of $\overline{H_\ell} = \overline{Q_\ell} \, \overline{K_{\ell-1}} \, \overline{Q_\ell}^\top$, therefore it is spsd. $D_\ell$ is a diagonal matrix that is just the concatenation of the diagonal parts of the local compressions, therefore it is also spsd. By induction, if $K_0 = K$, then all $K_\ell$ and $D_\ell$ matrices are spsd, and therefore the entire factorization (8) is spsd. ∎

**Proof of Proposition 2.** The cost of computing the compressions in a given stage is at most $b_{\max} s c_{\text{comp}} m_{\max}^{\alpha_{\text{comp}}}$. However, since $b_{\max} m_{\max} \leq n$, this is upper bounded by $s c_{\text{comp}} m_{\max}^{\alpha_{\text{comp}}-1} n$. The other component is the number of operations required to perform the rotations in each stage. The matrix $K_{\ell-1}$ has to be rotated from both the right and the left by $\overline{Q_\ell} = \bigoplus Q_i^\ell$, but since each row of these matrices is $c_{sp}$ sparse, the total per-stage complexity is bounded by $2c_{sp}n$. ∎

**Proof of Proposition 3.** The final core-diagonal matrix $H_s$ has a core of size $d_{\text{core}}$. Along with the remaining terms along the diagonal, this factor requires $d_{\text{core}}^2 + n - d_{\text{core}}$ storage. For each of the $s$ levels of the MKA, each factor $Q_\ell$ has $n$ rows, each of which is $c_{\text{sp}}$-sparse, so each of the $s$ factors has at most $c_{\text{sp}}n$ nonzero entries. Adding everything up yields an upper bound of $(sc_{\text{sp}}+1)n+d_{\text{core}}^2$. ∎

Table 1: Summary of the datasets used in our experiments

| Dataset | Size | Dimensions |
|---|---|---|
| housing | 506 | 13 |
| rupture | 2066 | 30 |
| wine | 4898 | 11 |
| pageblocks | 5473 | 10 |
| compAct | 8192 | 21 |
| pendigit | 10992 | 16 |

**Proof of Proposition 5.** As explained above each $Q_i^\ell$ from an MMF-compression is a product of at most $\lfloor \gamma m \rfloor$ Givens rotations, requiring $2\lfloor \gamma m \rfloor$ storage. All the $Q_i^\ell$ matrices in a given stage add up to $\sum_i 2\lfloor \gamma m_i^\ell \rfloor \leq 2n$ storage. The size of $H_s$ is the same as in Proposition 3. ∎

**Proof of Proposition 6 (sketch).** $\tilde{K}z$ is computed by multplying $z$ by each of the factors in (8), from right to left. Since each $Q_i^\ell$ is the product of at most $m$ Givens rotations, multiplying the corresponding block of a vector by $Q_i^\ell$ has complexity $2m$. Thus the complexity of multiplying a vector by $\overline{Q_\ell} = \bigoplus Q_i^\ell$ is at most $2n$. There are $s$ stages on the right of $H_s$ and $s$ stages on the left, leading to a bound of $2sn$. Multiplying a vector by $H$ itself has complexity at most $d_{\mathrm{core}}^2 + n$. ∎

**Proof of Proposition 7.** All of the matrix operations described in this procedure can boil down to computing a complete eigenvector decomposition (EVD) of $\tilde{K}$ and performing matrix operations on the resulting eigenvalues of the decomposition.

1. $\tilde{K}^\alpha = \sum_{i=1}^n \lambda_i^\alpha v_i v_i^T$ where $\{v_i\}_{i=1}^n$ is an orthonormal basis of $\tilde{K}$. Since $\tilde{K} = Q_1^T Q_2^T \cdots Q_s^T H Q_s \cdots Q_2 Q_1$, it suffices to compute an EVD of $H$ which is $d_{\mathrm{core}}$-core-diagonal. To compute the EVD it suffices to compute the EVD of $[H]_{[d_{\mathrm{core}}],[d_{\mathrm{core}}]}$, which requires $d_{\mathrm{core}}^3$ operations. Once the EVD is computed, to take the power of the eigenvalues requires only $n$ operations. All together, this is $O(n + d_{\mathrm{core}}^3)$ operations.

2. $\exp(\beta\tilde{K}) = \sum_{i=1}^n \exp(\beta\lambda_i) v_i v_i^T$ with notation as in $\tilde{K}^\alpha$. Again, the calculation of the EVD of $\tilde{K}$ costs $d_{\mathrm{core}}^3$ operations and the additional procedure of exponentiating $\beta$ times the eigenvalues takes $2n$ operations. Together this costs $O(n + d_{\mathrm{core}}^3)$ operations.

3. $\det(\tilde{K}) = \prod_{i=1}^n \lambda_i$. Every rotation matrix has a determinant equal to one, so the $Q_l$ terms which are block-rotation matrices, will also have determinant equal to one. Computing the determinant of $H$ again boils down to computing the EVD and then taking the product of the eigenvalues. This will also have $O(n + d_{\mathrm{core}}^3)$. ∎

# 3 Algorithm

The pseudocode of the proposed **Multiresolution Kernel Approximation (MKA)** algorithm is shown in Algorithm 1. MKA is a meta-algorithm, in the sense that it can be used in conjunction with different core-diagonal compressors.

# 4 Experiments

## 4.1 Datasets

We used six data sets in our experiments, of which, `rupture` is from Materials algorithms project program [1] and the others are from UCI machine learning repository [2]. The detailed summary of the datasets is in Table 1.

**Algorithm 1** The MKA algorithm. COMPRESS is any suitable core/diagonal compression routine, e.g., a Jacobi MMF.

---

**Input:** an spsd kernel matrix $K \in \mathbb{R}^{n \times n}$
$K_0 \leftarrow K$
**for** $(\ell = 1 \text{ to } s)\,\{$
    **cluster** the columns of $K_{\ell-1}$ into $(\mathcal{C}_1^\ell, \ldots, \mathcal{C}_{p_\ell}^\ell)$
    **permute** the rows/columns of $K_{\ell-1}$ according to $(\mathcal{C}_1^\ell, \ldots, \mathcal{C}_{p_\ell}^\ell)$ to get $\overline{K_{\ell-1}}$
    **for** $(i = 1 \text{ to } p_\ell)\;\{$
        $(Q_i^\ell, c_i^\ell) \leftarrow \text{COMPRESS}([\overline{K}_{\ell-1}]_{i,i})$
    $\}$
    $\overline{Q}_\ell \leftarrow \bigoplus_i Q_i^\ell$
    $\overline{H}_\ell \leftarrow \overline{Q}_\ell \overline{K}_{\ell-1} \overline{Q}_\ell^\top$
    $c_\ell = \sum_{i=1}^{p_i} c_i^\ell$
    **permute** the rows/columns of $\overline{H}_\ell$ so that the cores appear in the top left $c_\ell \times c_\ell$ submatrix to get $H_\ell$
    $K_\ell \leftarrow [H_\ell]_{1:c_\ell, 1:c_\ell}$            // this is the "core" part of $H_\ell$
    $D_\ell \leftarrow \text{diag}(\text{diag}([H_\ell]_{c_\ell+1:, c_\ell+1:}))$      // this is the "diagonal" part of $H_\ell$
$\}$
**Output:** $(\overline{Q_1}, \ldots, \overline{Q_s}, D_1, D_2, \ldots, D_s, K_s)$

---

Figure 2: SMSE and MNLP as a function of the number of pseudo-inputs/$d_{\text{core}}$ on the rest four datasets. In the given range MKA clearly outperforms the other methods in both error measures.

## 4.2   Evaluation as a function of number of pseudo-inputs/$d_{\text{core}}$

We compare regression results in terms of both the predictive mean and variance (i.e. SMSE/MNLP) as a function of the number of pseudo-inputs/ $d_{\text{core}}$, which represents the approximation/compression level of the kernel matrix. Across the range of pseudo-inputs/$d_{\text{core}}$ considered in Figure 2 for selected data sets in this supplementary material , MKA outperformed other methods in terms of both prediction accuracy (SMSE) and variance assessment (MNLP), whereas for other methods more error was accumulated as fewer pseudo-inputs were used. These results on additional data sets to those illustrated in the main paper confirm the position that MKA is in many cases a superior method for kernel matrix compression. In these four data sets, as well, MKA's performance was nearly constant across different sizes of $d_{\text{core}}$ – likely due to the information preserved along the main diagonal in the $c$-core diagonal matrix of MKAs kernel matrix approximation. Moreover, the results for MEKA on the selected data sets are absent due to the fact that the approximate kernel matrix found by MEKA for these data sets loses the spsd property, and thus fails to show prediction results in the experiments.

## Footnotes

[1] https://www.phase-trans.msm.cam.ac.uk/map/map.html

[2] https://archive.ics.uci.edu/ml/datasets.html