[Reviews · NeurIPS 2017]

Reviewer 1



The authors consider the problem of large-scale GP regression; they propose a multiresolution approximation method for the Gram matrix K. In the literature, most approximation approaches assume either (1) a low rank representation for K, which may not be supported by the data, or (2) a block-diagonal form for K, the structure of which has to be identified by clustering methods, which is not trivial for high-dimensional data. The current paper proposes MKA, a novel approximation approach that uses captures local and global properties for K. The Gram matrix K is approximated as a Kronecker sum of low-rank and diagonal matrices, a fact that significantly reduces the computational complexity of the linear algebra calculations required in the context of GP regression. The paper initiates a very interesting discussion on the nature of local and global kernel approximations, but I feel that certain aspects ofthe methodology proposed are not sufficiently clear. Below, I list some considerations that I had while reading the paper. What is the effect of the clustering mentioned in step 1 of the methodology? Is the method less sensitive to the result of clustering than the local-based methods in the literature? Does the approximate matrix \tilde{K} converge to true matrix K as the d_{core} parameter is increased? By looking at the experiments of Figure 2, it appears that MKA is rather insensitive to the d_{core} value. The effect of approximating K in many stages as described in Section 3 is not obvious or trivial. The authors attribute the flexibility of MKA to the reclustering of K_l before every stage. However, I would suspect that a slight variation of the output at any stage would have dramatic consequences in the stages that follow. It is not clear which compression method was used in the experiments. I would think that the cost of SPCA is prohibitive, given the objective of the current paper. It could still be worth mentioning SPCA if there was some experimental comparison with the use of MMF in the context of MKA. I think that the experimental section would be stronger if there was also a demonstration of how well the MKA approximates the original GP. Although we get an idea of the predictive mean in Figure 1, there is no information of the predictive variance. Minor comments: The line for the Full GP method in Figure 2 is almost not visible.

Reviewer 2



The paper introduces a new kernel approximation method enabling inversion of positive symmetric matrices with linear complexity. In contrast to previous methods, it is not based on a low rank approximation but instead uses local factorization on a so called c-core-diagonal form with a hierarchical factorization. The idea is interesting and the results are very promising. Nevertheless, I am lacking some more intuitive discussion why this approach is superior to other methods in the comparison. As I understand it there are two novel components in this paper. 1. The hierarchical factorization and 2. the factorization into a c-core-diagonal form. The authors have not fully explained why these ingredients are important. Also, can these two strategies be used separately or only combined? I am also lacking details such as: * Based on what criteria are the rows/columns in K_0 clustered? * How are the number of clusters p_l decided? * How do you decide the size of c and the corresponding indices CD(c)? Also, the presentation of the algorithm could be clearer. This would be improved if you move the pseudo algorithm in the supplementary material to the main paper I think. Minors: * The notation K' appears after eq (2) but never introduced (I assume K'=K+sigma^2 I ) * Row 248: "than than" -> "than"

Reviewer 3



The paper applies a fast hierarchical matrix decomposition to the task for inverting and finding the determinant of the kernel matrix of kernel based methods. The overall complexity of the algorithm is equivalent to or lower then calculating the values in K, O(n^2). In contrast to inducing point based methods, here the kernel is decomposed in such a way as to capture both the long range (dubbed "PCA-like") and short-range (dubbed "k-nearest neighbour type") interactions in data. The paper is clearly written and gives good intuitions as why the approach works, a good description of the algorithm and compelling results on real data.